# Real-Time Detection of Faults in Rotating Blades Using Frequency Response Function Analysis

Ravi Prakash Babu Kocharla [1] , Murahari Kolli [2] and Muralimohan Cheepu [3,4,*]

1 Department of Mechanical Engineering, Prasad V. Potluri Siddhartha Institute of Technology, Vijayawada 520007, India
2 Department of Mechanical Engineering, Lakireddy Bali Reddy College of Engineering, Mylavaram 521230, India
3 Department of Materials System Engineering, Pukyong National University, Busan 48547, Republic of Korea
4 STARWELDS Inc., Busan 46722, Republic of Korea
* Correspondence: muralicheepu@gmail.com; Tel.: +82-10-2714-3774

**Abstract:** Turbo machines develop faults in the rotating blades during operation in undesirable conditions. Such faults in the rotating blades are fatigue cracks, mechanical looseness, imbalance, misalignment, etc. Therefore, it is crucial that the blade faults should be detected and diagnosed in order to minimize the severe damage of such machines. In this paper, vibration analysis of the rotating blades is conducted using an experimental laboratory setup in order to develop a methodology to detect faults in the rotating blades. The faults considered for the study include cracks and mechanical looseness for which dynamic responses are recorded using a laser vibrometer. Analysis has been carried out by comparing the frequency response function spectrums of the fault blade with those of the healthy blade related to the resonance frequency. The Internet of Things and wireless sensor networks are implemented to transmit the measured data to the cloud platform. A support vector machine algorithm is used for preparing the learning model in order to extract and classify the faults of the rotating blades. It can be clearly seen from the results that there is variation in the frequency response function spectrums of healthy and faulty conditions of the rotating blades.

**Keywords:** fault detection; cracks; mechanical looseness; vibration spectrums

## 1. Introduction

Fault detection and diagnosis are concepts that have existed from the inception of machines. In the early days, the view of the manufacturers and the users towards fault diagnosis was that the safety and protection of a machine ensured its reliable operation. However, nowadays machines are expected to perform multiple tasks which make their operations more complex. Therefore, there comes the requirement to improve the fault diagnosis methods.

Rotating blades are important components in machines such as turbines, fans, pumps, etc., used in modern mechanical industries. These blades convert some form of energy into mechanical power. For example, wind turbines convert wind energy, when passed over the blades, into mechanical power by developing aerodynamic force. Likewise, steam turbine blades make the shaft to rotate when high pressure steam passes over them. Additionally, equipment such as draught fans, blowers and compressors make use of the kinetic energy of blades in order to increase the fluid outlet pressure. Such blades can become damaged during critical operating conditions and have adverse effects on the safety and reliability of the machines. Therefore, it becomes evident that developing a condition monitoring method that can detect the faults during blade operation is necessary. Every fault in the rotating blades may cause unwanted vibrations in the machine and carry the information of the fault. Therefore, vibration-based condition monitoring of the blades from the time of their inception can help us to detect blade faults at the earliest.

Sometimes, rotating blades also need to be monitored for defects developed during the manufacturing process or made by less qualified personnel. Therefore, quality management is essential during product development along with condition monitoring during the performance stage of the blades. Any methodology developed for condition monitoring can be used for quality assurance, along with the quality control of the blades. Emerging technologies such as the Internet of Things and machine learning techniques, along with cloud computing, can be implemented in developing real time fault detection methods that can achieve efficient quality management of the blades. In doing so, sensors play a vital role in measuring and acquiring data from the rotating blades. Pennacchi and Vania [1] have conducted experiments on a flexible coupling connecting two shafts in order to analyze the effect of angular misalignment. It is concluded that the vibration-based condition monitoring of a rotor can be simulated to predict faults in rotating machines. Djaidir et al. [2] have analyzed the dynamic behavior of gas turbine rotors with defects using mathematical techniques of vibration response. In order to validate the usefulness of the mathematical models, experiments are conducted and it has been found that the developed approach is very useful in the fault diagnosis of gas turbines. Guan et al. [3] have determined the rotational frequency from the derived vibration equations of misalignment states (i.e., angular and offset) through shaft dynamic models and verified the results with simulation and experimental results. Al Adawi and Ramesh Kumar [4] have investigated the cause of high vibrations developed in the gas turbine, generator rotor and gear box of a power station. This could help to detect the fault causing the vibration and prevent any chance of catastrophic failure of the components.

Chaun Li et al. [5] investigated the influence of different domains of deep learning algorithms in order to improve fault diagnosis capability. Andre et al. [6] developed a condition monitoring system which is based upon cloud computing that includes data mining and machine learning techniques. Failures in rotating machines are identified by utilizing the data available in the real time Ethernet network, which will reduce the number of sensors. This helps in reducing the maintenance cost of sensors, investment towards signal processing hardware and developing a user-friendly system that can be understand by the industry personnel. Marwin et al. [7] evaluated real time data using a learned model for the predictive maintenance of a multipurpose machine. The clustering approach is used for the segmentation of raw data, and then data are processed for predicting the degradation state of the machine using a machine learning classifier. Wang et al. [8] have proposed a new fault detection method based on a convolutional neural network (CNN) using bottle neck optimization of multi-vibration signals of a wind power test rig. The multi-vibration signals obtained from the sensors are first converted into 2D images and then features are extracted using CNN technique, and then the faults are detected.

Min Xia et al. [9] proposed a framework in which IoT and cloud computing can be assisted in the continuous development of a design in order to improve its efficiency and cost effectiveness. Siliang Lu et al. [10] proposed a condition monitoring method in order to diagnose the faults in motor bearings. This method works on the undersampling principle as it reduces the sampling rate, data transmission time and power consumption while measuring and collecting the different signatures of the vibration signals of the defective bearings. Adrian Stetco et al. [11] presented a review on condition monitoring of wind turbines for the detection of blade faults using machine learning models. It was proposed that support vector machines, neural networks and decision trees are the most suited machine learning algorithms in order to select and extract the features of the fault.

Andreas Theissler et al. [12] presented an overview of the most suitable machine learning techniques used for the predictive maintenance of automotive applications. Yoonjae Lee et al. [13] proposed a feature variable dimensional coordination method for fault diagnosis that can result in high accuracy with less processing time. Mohanta et al. [14] reviewed the detection of several sources of faults such as imbalance, bearing problems, wicket gate problems and shear pin failure in hydro power stations by monitoring vibration signals of turbine-generator sets. Chenyang Li et al. [15] proposed a condition monitoring system that

utilizes the vibration data of machinery equipment measured and transmitted by a terminal device based upon the IoT. Using these data, a diagnosis model was trained that can detect the imbalance caused by a blade in a fan. Alexandre et al. [16] evaluated the performance of various machine learning algorithms by acquiring data during predictive maintenance in order to diagnose the rotating machines. In recent years, the condition monitoring of industrial equipment has been carried out by making use of the concepts available in emerging technologies such as the Industrial Internet of Things (IIoT) and the Fourth Industrial Revolution. The sensors capable of acquiring the data in the form of images, graphics and digitalization are processed to identify the condition of the equipment [17].

Omar et al. [18] presented a review of the challenges facing the condition monitoring and fault diagnosis of induction motors using acoustic emissions. This review was focused mainly on the faults in bearings, rotor and stator. Shiqing et al. [19] investigated the capability of trumpet-shaped acoustic metamaterials with a high refractive index in order to magnify the specific frequency range of the acoustic signals generated during the operation of machines. Experimental studies of such metamaterials have proved to have more efficient sensing with high performance when diagnosing faults in machinery. Divya et al. [20] proposed an intelligent fault detection system to detect faults in an air compressor. The system is based on the real-time empirical acoustic sensor data acquired from the key sensitive points that are identified by performing sensitive point or position analysis. The faulty features are determined using a wavelet-based approach, and a model-based supervised classifier is employed to predict the fault class. Lu et al. [21] evaluated the condition monitoring performance of a two-stage helical gearbox using airborne acoustics in conjunction with modulation signal bispectrum analysis. Qurthobi et al. [22] conducted an evidence-based systematic review on the different approaches and techniques used for mechanical failure analysis based on acoustic signals. Caleb et al. [23] proposed a fault diagnosis method for drills based on principal component analysis and artificial neural networks. The sound data are collected for the undamaged and defective drills, and further processed to identify the faults. Martin-del-Campo and Sandin [24] presented the measures and heuristics for the automatic scoring of the acoustic emissions developed due to faulty condition in the bearings of a rotating machine.

Saeed et al. [25] proposed a fault diagnosis approach in rotating equipment based on permutation entropy, signal processing and artificial intelligence. Permutation entropy is implemented to identify the faulty state from the extracted frequency components using wavelet packet transform and envelope analysis of the vibration signals. The multi-output adaptive neuro-fuzzy interface system is implemented to classify the fault type. Scislo [26] introduced a vibration measurement system that works on a completely non-contact approach, which includes sound pressure change as the excitation source and the 3D Laser Doppler Vibrometer as the measurement system. The Industry 4.0 data acquisition and analysis systems enabled the development of quality assurance and control of the production process. On the other hand, the fabrication method of the blades and wheels also might have an effect on the performance and quality aspects [27–32]. During fabrication or manufacturing of the objects the induced stresses, heat and powder forms of additive methods has an effect even at micron levels, which are sensitive to centrifugal forces and torque [33–37]. Brito et al. [38] proposed a new approach for fault detection and diagnosis in rotating machines in which an unsupervised anomaly detection algorithm was used to identify and classify the faults through vibration analysis. Root cause analysis was performed when unsupervised classification was not possible to classify the faults.

It can be clearly understood from the literature that the power generating machinery uses blades that may fail during operation under critical loading conditions. Therefore, there exists a need to develop a condition monitoring technique in order to diagnose blade faults. The aim of this paper is to detect the faults of the rotating blades using vibration-based condition monitoring. In order to achieve this, a rotating blade test rig was designed and manufactured for healthy and faulty conditions. A laser beam vibrometer was used to measure the vibration signals of the rotating blades and then processed through a Fast

Fourier Transform (FFT) analyzer in order to extract the frequency response function (FRF) spectrums. The support vector machine learning technique was used in order to classify the faults based on the extracted features of the rotating blades. The work presented in this paper is believed to be useful in finding the blade faults of machinery operating in remote places with minimum human intervention.

## 2. Materials and Methods

In certain applications of rotating blades, such as thermal power plants and aerospace jet engines, turbine blades are subjected to high temperatures along with centrifugal loads. However, as these blades are coated with thermal resistant coatings, they are able to withstand thermal loads. Additionally, the effect of the elevated temperatures on the blades is appreciable only when the blade material has crystalline defects such as dislocations. Therefore, temperature environment is not considered in this work and it is assumed that faults are developed purely due to mechanical loads.

An experimental rig is designed in the resemblance of rotating blades in order to develop a health monitoring system to detect the blade faults. A laser beam vibrometer is used to measure the vibration response so that investigations can be carried out by comparing measured FRF spectrums of the healthy and faulty blades. The IoT helps the measured data to be transmitted to the cloud platform, wherein the classification of the fault is achieved by applying the suitable machine learning technique.

### 2.1. Manufacturing of the Experimental Rig along with Instrumentation

For this research study, a test rig was manufactured in accordance with modelled dimensions as shown in Figure 1. Finite element based modal analysis was carried out using ANSYS R15.0 software to determine the natural frequency of the rotating blades. To validate the theoretical results obtained from the ANSYS software, experiments were conducted on the test rig. Knowing both the theoretical and experimental results with less deviation, it was decided to carry out further analysis using the test rig as it can be justified with real time operating conditions. The blade faults, such as cracks and mechanical looseness, were developed within the test rig for which vibration signals are measured in order to develop the blade health monitoring system.

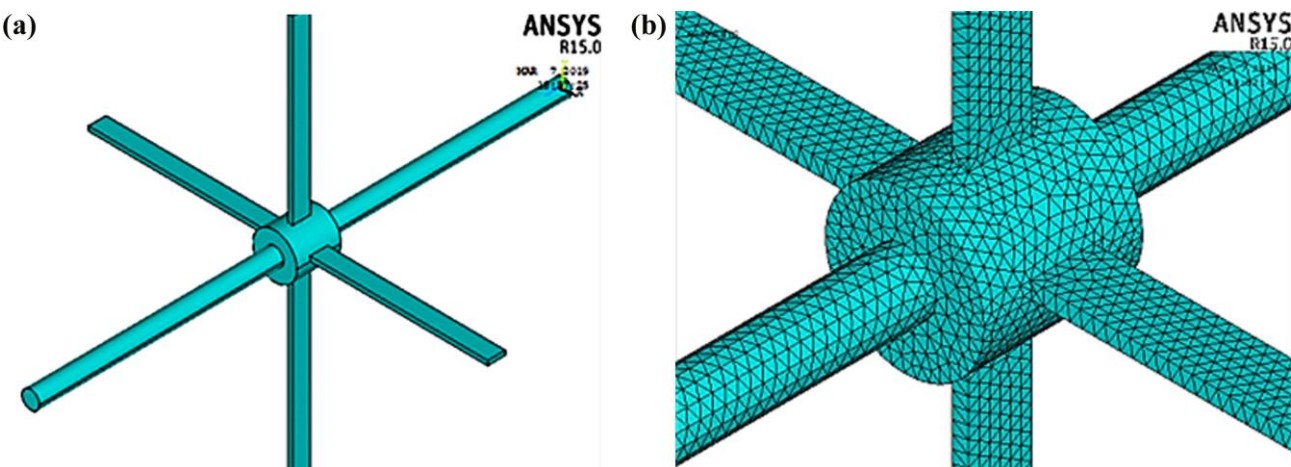

**Figure 1.** (**a**) Geometric and (**b**) finite element models of the shaft.

The boundary condition for the rotating blades is shown in Figure 2. For shell 63 (4 node shell element), every node has 6 degrees of freedom, 3 translations, Ux, Uy and Uz, and 3 rotations: Rotx, Roty and Rotz. Ux, and Uy degrees of freedom of the nodes, which are at supports, are constrained to simulate the actual condition of rotating blades.

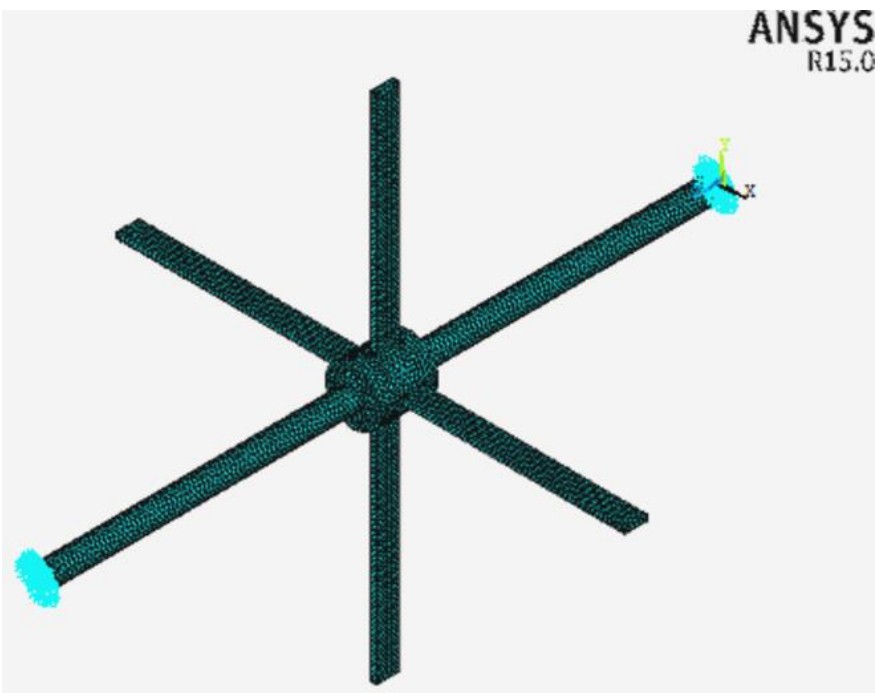

**Figure 2.** Boundary conditions of the rotating blades.

The experiments were conducted using a set-up as shown in Figure 3a, comprising of shaft, coupling, hub, blades, motor, etc. FRF signals of vibration are measured by laser vibrometer, as shown in Figure 3b. In order to fetch the acceleration signal of vibration, a laser was focused on the rotating shaft driven by a motor. These signals were obtained for each case, i.e., healthy, crack and mechanical looseness. The blade faults, such as cracks and mechanical looseness, were developed within the test rig for which vibration signals were measured in order to develop the blade health monitoring system. FRF spectrums were extracted, and then comparisons were made between healthy and fault conditions.

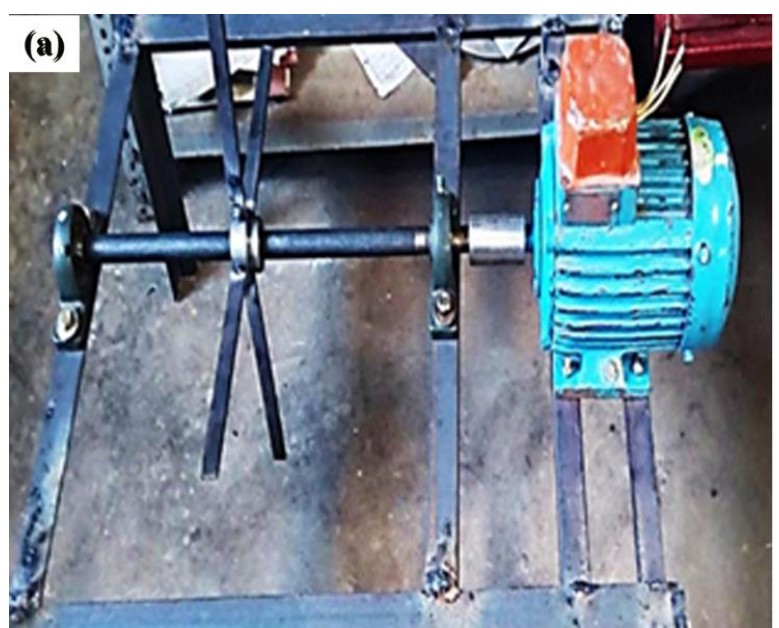
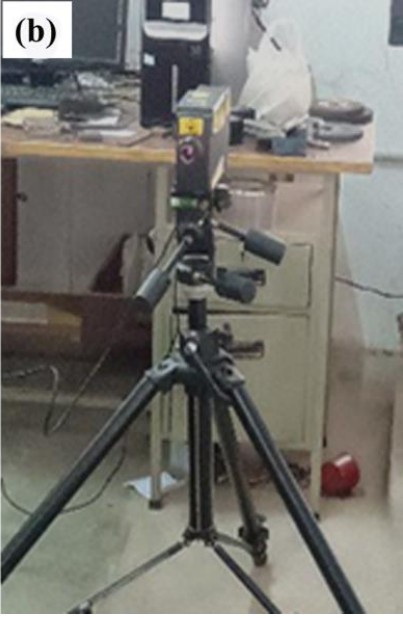

**Figure 3.** (**a**) Experimental rig and (**b**) laser vibrometer.

The detailed discussion of the experiment set-up is as follows. A shaft of 20 mm diameter and 600 mm length, made of steel, was supported by two ball bearings at its ends. The blades were attached to a hub, which was centrally located on the shaft. This entire assembly was connected to the motor with a flexible coupling designed to transmit torque without any axial, radial or angular misalignment. The rotational motion of the shaft was attained by a power-driven motor of Kirloskar make of 1 HP and 3000 rpm speed. The shaft-motor assembly was mounted on a supporting frame in alignment with the shaft center using bolts and nuts.

Frequency response function spectrums are the measurement of vibration signals of the rotating blades. A laser vibrometer is a non-contact type vibration measuring instrument and is used to measure the frequency spectrum of the rotating blades. A Fast Fourier Transform analyzer is connected to the vibrometer in order to convert the vibration signal into a convenient form which can be later assessed for detecting the faults of the rotating blades. The experimental work of this research was conducted on the test rig, which is located in the Machine Dynamics laboratory at PVPSIT, India.

Non-contact type sensors can reduce the use of a greater number of contact sensors to measure the detailed vibration response of a rotating blade. Non-contact type sensors such as laser vibrometers are available in 1D or 3D based upon the detailed vibration measurement. The 3D laser vibrometers provide more detailed vibration information compared to other vibrometers. Therefore, the selection of the vibrometer depends upon the amount of vibration details. For the fault detection of rotating blades, 1D vibrometer information is sufficient to perform frequency response function analysis.

The laser vibrometer is of Polytec make, consisting of an OFV511 compact sensor head along with a laser unit, as shown in Figure 3b. This laser unit delivers a beam of helium neon laser light of 316 nm through an optical fiber and is received by a high precision interferometer available in the head. The laser beam is split by a beam splitter into two beams, namely a reference beam and a measurement beam. The measurement beam is then focused onto the rotating blades by making it pass through a second beam splitter. The reflected measurement beam is then merged with the reference beam by the third beam splitter and focused on the photo detector. A Doppler frequency shift of the measurement beam is manifested with the change in the optical path length per unit of time. An acousto-optic modulator placed in the reference beam creates the frequency shift, and the modulation frequency is directly proportional to the velocity of the rotating blades.

### 2.2. Modal Testing

The process of determining the modal characteristics is referred to as modal analysis. The modal characteristics include natural frequencies, mode shapes, stiffness and damping of the vibrating component. The knowledge of these modal characteristics of any component helps to predict its dynamic behavior. The modal characteristics contain the most vital information in the design of rotating machinery. They are required for carrying out the dynamic analysis of components in the finite element environment by knowing vibration response. The vibration responses of the components are then extracted from modal testing in the form of FRFs to obtain natural frequencies. The known natural frequencies are helpful to avoid resonance during operation, and in turn help to elevate the operational life of the components. Additionally, the information of the vibration response is utilized to detect the faults in the components with reference to the vibration response of the healthy component.

The natural frequencies of the rotating blades are determined by conducting the modal testing by using the FRF test. The experimental rig for the modal testing of the rotating blades is shown in Figure 4. The laser vibrometer measures the vibration responses, which are analyzed using FFT and recorded on a computer. From the FRFs of the accelerated blade, natural frequencies are identified.

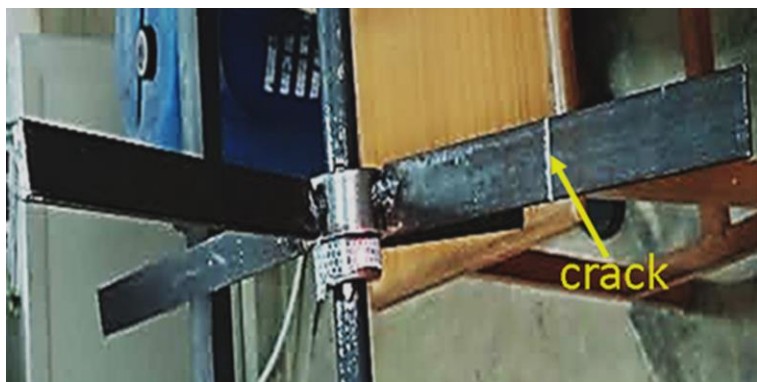

**Figure 4.** Cracked blade.

*2.3. Online Condition Monitoring*

The main aim of the condition monitoring is to prevent machines from experiencing catastrophic failure by the early detection of the faults. This helps to diagnose the faults and eliminate further damage to the machine. Therefore, an effective structural health monitoring system is needed for the detection and diagnosis of the faults in real time industrial applications. In this section, a brief description of the proposed condition monitoring system based on real time online technology is given for fault detection and diagnosis. Wireless sensor networks (WSNs) and the IoT are utilized in the proposed methodology, as is shown in Figure 5. The entire system design comprises the disposition of the WSNs, monitoring place and cloud platform.

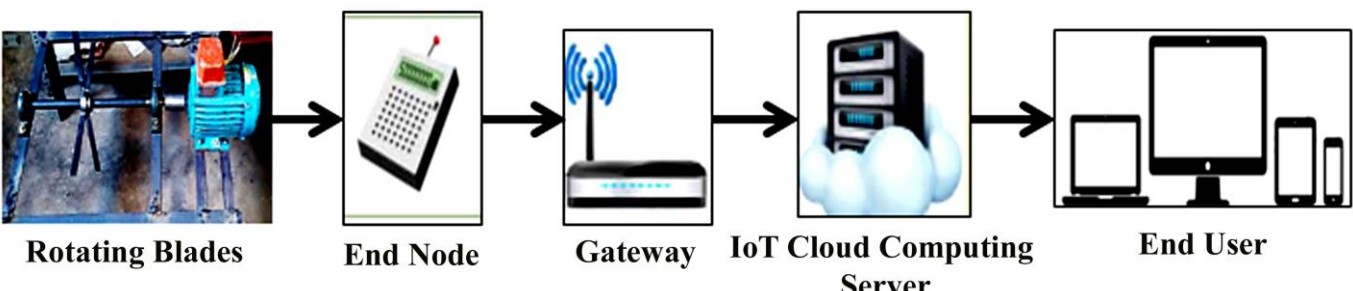

**Figure 5.** Schematic diagram of the online condition monitoring system.

2.3.1. Disposition of the WSN System

In applications evolving for automation, attention is being grabbed globally by wireless communication technologies. One such standard adopted in the proposed system is IEEE 803.15.4. Figure 6 shows the functional diagram of the WSNs system proposed in this work. The measurements of the condition monitoring parameters such as vibration and natural frequencies are obtained by using a vibration sensor focused on the rotating blades of the machine. When several sensor nodes are used for fault detection in more than one machine, it is required that one of the sensor nodes becomes the basic node, which helps to handle the allocation of the resources, blending of the data and scheduling of the network seeing the prior condition of data transmission. This functionality implementation prevents unnecessary data and decreases the flow of data in the wireless communication network, thus increasing the life of the sensors. Sensors send the vibration information of all the machines to the basic node and then the data are transmitted in the form of packets to the monitoring place. Then, the interface of the coordinator node with the expert system is carried out through serial port communication.

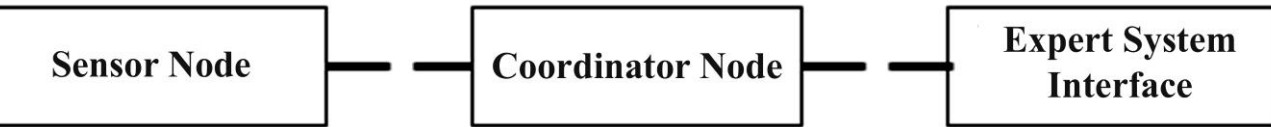

**Figure 6.** Data transmission through IoT.

### 2.3.2. Fault Diagnosis Method in Rotating Blades

The sensor used in this work is a laser vibrometer. Its function is to measure the vibration signals of the healthy and faulty conditions in the blades. These signals will help in understanding the frequency response function spectrum for each condition of the blades, so as to detect faults in real time applications. This system is used to process the data acquired from the sensors and helps to detect and diagnose the faults in the machines. When the spectrums extracted by this system increase, it becomes difficult to classify the detected faults and it is necessary to apply the dimensionality set of the spectrum. Therefore, it is proposed to classify the faults by implementing machine learning techniques such as support vector machines, neural networks, decision trees, etc. This process of fault detection is divided into three phases, namely preprocessor, feature extraction of spectrums and finally classification of the fault and its diagnosis.

The vibration response data of the rotating machine are collected and analyzed in preprocessor stage. These analyzed data are purely mechanically based and help to detect the faults when processed in real time. The data processing includes the removal of the weak data by the host system that are entered during data acquisition by WSNs. The faults in the rotating machine can be either individual or combinational. In order to differentiate the type of the fault, the spectrums are extracted in the feature extraction stage. The support vector machine algorithm is proposed in order to analyze the spectrums obtained for various cases of faults in the machine. From the results obtained from the analysis, the type of the fault is listed and the severity of each fault is identified. In the last phase, the extracted features are compared in order to classify the type of the fault in the rotating blades of a machine.

### 2.3.3. Cloud Platform of IoT

The method used for the extraction of features from FRF spectrums used in the detection of faults is found to be most suited method and accurate enough in classifying the fault. The support vector machine algorithm is more relevant for fault prediction in real time applications using the cloud platform of the IoT. For performing such IoT operations, it is required to choose a cloud service from the vendors. Services provided by the cloud vendors include remote control, data analysis, machine learning, deep learning, etc., depending upon the requirements of the application.

It is necessary to store the data and analyze it in real time. Therefore, a cloud platform is selected, namely ThingSpeak. In order to implement ThingSpeak in this research work, the foundation is the instance creation of the parameter that serves as the interconnection between the IoT and data storage services. Additionally, a programming tool that links the real time application environment is also provided by ThingSpeak. Python code is developed for the extraction of the features and for classifying them for detecting the faults. This code receives data from the sensors through the main functional block as they are linked together with the IoT. The data sent by the sensors will be in the form of a message, and need to be extracted before being processed by the Python code. Due to the service being provided by the IoT, communication happens among the real time application and sensors via cloud platform. All the devices in the IoT are interlinked and communicate among themselves with the aid of a lightweight protocol. Therefore, in the work of fault detection and diagnosis using the IoT, the transmission of data is carried out in the form of lightweight protocol or messages.

In the proposed condition monitoring system of fault detection, a support vector machine algorithm helps to process the vibration data transmitted by the IoT network,

as shown in Figure 7. The efficiency of the fault classification can be improved by comparing low and high ranges in feature extraction, taking into account accuracy, precision and specificity.

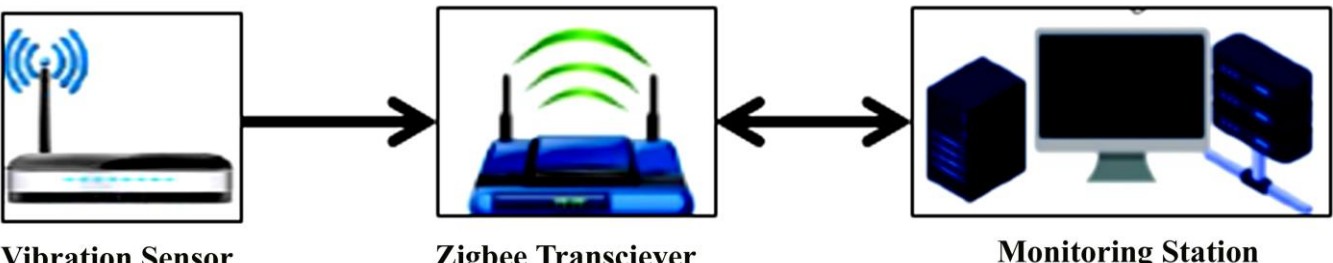

**Figure 7.** Actual set-up of nodes in IoT.

## 3. Results and Discussions

The first three resonance frequencies of the rotating blades for both healthy and faulty conditions were evaluated using finite element analysis software and are listed in Table 1, and the corresponding mode shapes are shown in Figure 8.

**Table 1.** Resonance frequency values of rotating blades.

| Condition | 1st Mode Frequency (Hz) | 2nd Mode Frequency (Hz) | 3rd Mode Frequency (Hz) |
|---|---|---|---|
| Healthy | 77.01 | 599.04 | 893.71 |
| Single crack | 74.78 | 596.8 | 886.2 |
| Two cracks | 74.92 | 594.6 | 878.1 |
| Looseness | 73.24 | 593.2 | 873.5 |

The natural frequencies corresponding to the first mode shape of the rotating blades as shown in Figure 8a are 77.01 Hz, 74.78 Hz, 74.92 Hz and 73.24 Hz for healthy, single crack, two cracks and mechanical looseness, respectively. It can be clearly seen that the natural frequencies of the first mode shape are different for the healthy and faulty conditions of the rotating blades, and also a significant decrease in the frequency value for all the faulty conditions compared with that of the healthy condition of the blades is observed. The same trend can be seen for the natural frequencies given in Table 1 of the second and third mode shapes of the rotating blades, as shown in Figure 8b,c. Therefore, it is evident that the decrease in the natural frequency of the rotating blades indicates the presence of a fault in one form or other.

The abnormal condition of the blade is found by using the Campbell diagram, from which the critical speed of the blades was identified, comparing the excitation and natural frequencies of the rotating blades. Whenever the rotating blades operate at the resonance condition, crack will initiate due to the higher stresses developed at the stress concentrated region of the blade. These stresses keep on increasing and the crack propagates until the point where the stress intensity factor of the crack exceeds the fracture toughness of the blade material before catastrophic failure occurs [39,40] That is why stress intensity factors are to be estimated for the propagating crack lengths based upon the linear elastic fracture mechanics. For the above condition of the failure, it is seen in the literature that the final crack length is estimated to be 70% of the width of the blade, which means that the blade does not fail at the crack length below final crack length. Therefore, cracks should be detected before they reach final length.

Vibrations are produced in the blades due to the faults, namely crack, mechanical looseness, imbalance, misalignment, etc. That means that vibration data can be used to detect such faults. Generally, faults such as imbalance and misalignment are monitored regularly during preventive maintenance works, but the cracks and mechanical looseness of the blades becomes a challenging task to detect. Therefore, it becomes necessary to

monitor the condition of the blades in order to detect cracks and mechanical looseness during the operation of the machine.

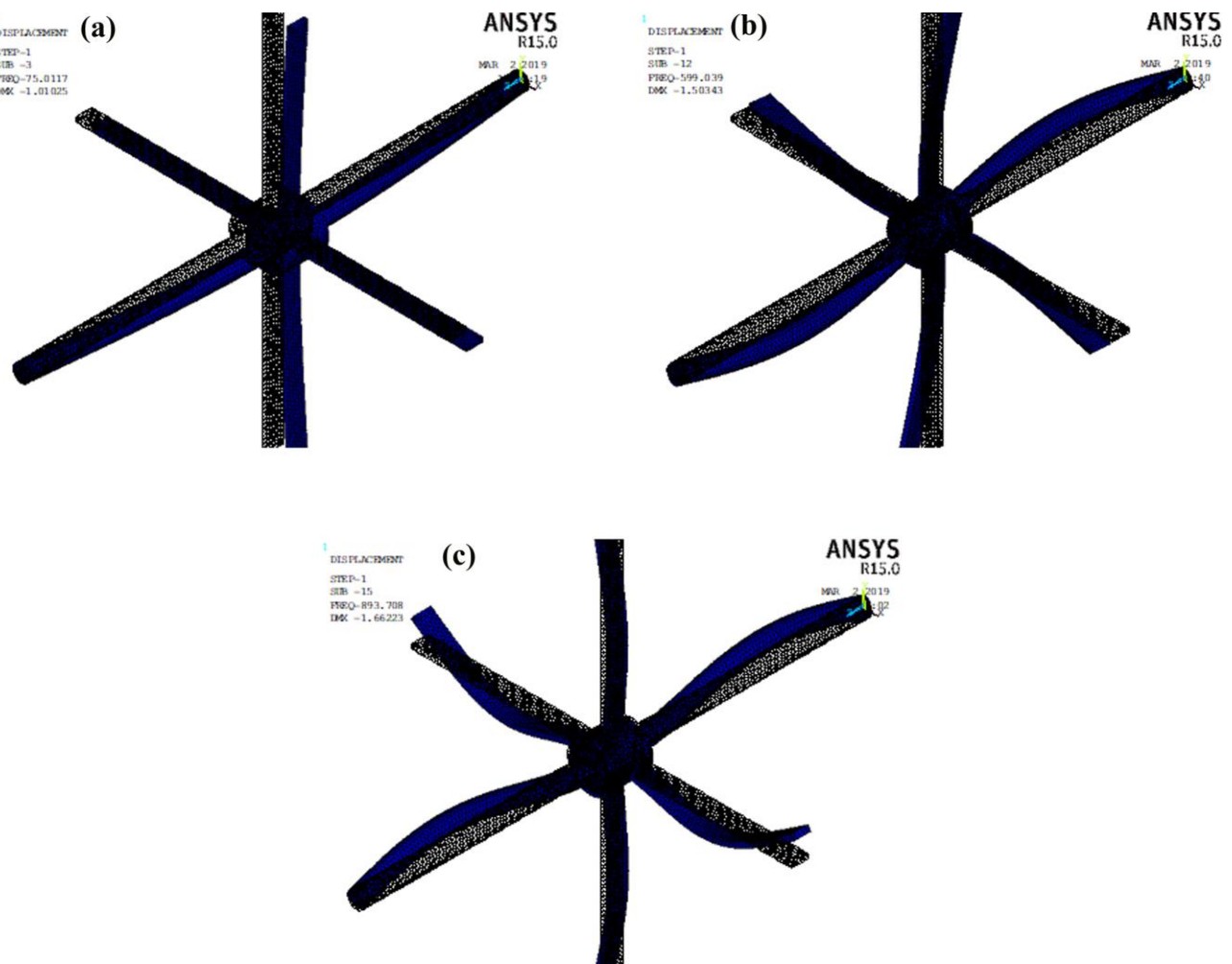

**Figure 8.** (**a**) First mode, (**b**) second mode and (**c**) third mode of the rotating blades.

Experiments are conducted on a rotating blades rig for two different blade faults, i.e., crack and mechanical looseness. These faults are selected based up on the actual facts observed in real time applications of rotating blades. The acceleration data are extracted for the rotating blades by conducting modal tests which are used to assess the health condition. This gives an understanding of how to investigate the effect of the faults on the blades, such as crack and mechanical looseness which are observed quite frequently in rotating blades. In order to implement the present work, one sensor node is utilized to measure the vibration of the rotating blades. The vibration data are transmitted to the monitor station using an IEEE 802.15.4 wireless communication device. The received vibration signal is the waveform in time domain that consists of statistical features. As these time domain features are difficult to use when classifying the faults, they are transformed into frequency signals and the frequency band is identified.

Measured signals of vibration are indicated in the form of harmonics for healthy and faulty conditions of the rotating blades. The FRF of a healthy blade is shown in Figure 9, and it is clearly seen that there is a higher harmonic at 77 Hz, which indicates that the resonance of the blades exist within that range and that blades may become excited during turbine start up or shut down at this fundamental natural frequency. The first fault condition considered is a crack on one of the rotating blades. Figure 10 shows the FRF spectrum of the rotating blades, and it is observed that the resonance frequency shows more peaks with

different amplitudes instead of the single peak seen for healthy blades. Additionally, the natural frequency of the cracked blade is reduced compared with the healthy blade. The second fault condition considered for the rotating blades was cracks on two blades which are adjacent to each other. The FRF spectrum indicates a greater number of peaks with increased resonance frequency range, as shown in Figure 11. The third fault condition for the rotating blades is the mechanical looseness of the blade for which the FRF spectrum has multiple peaks, especially on lower side of resonance frequency range of the blade, as shown in Figure 12.

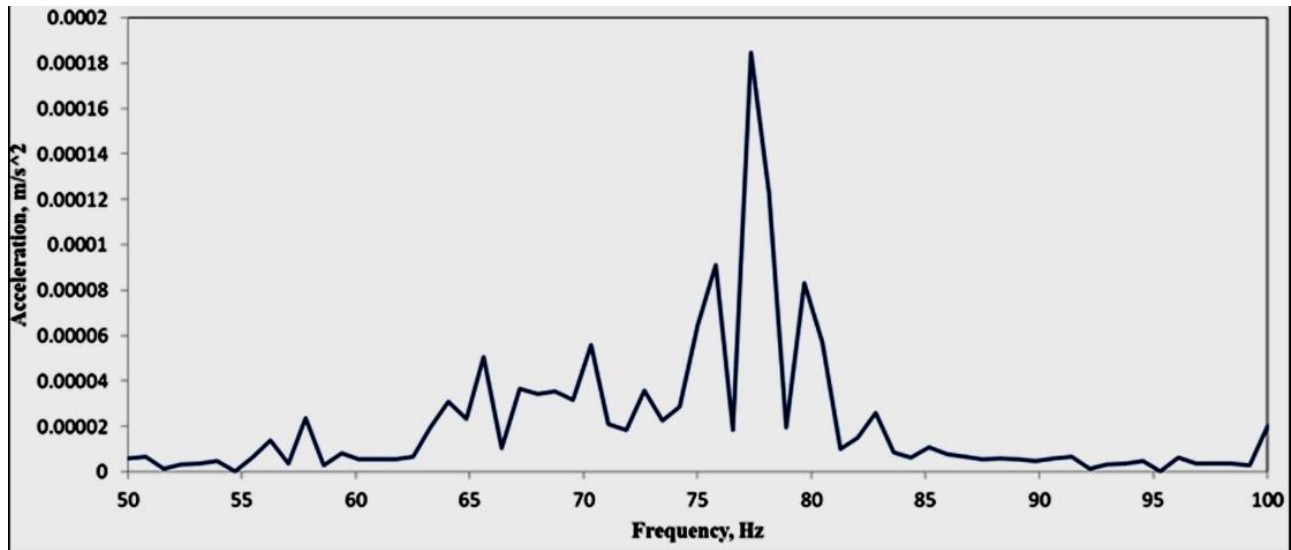

**Figure 9.** FRF of Healthy blade.

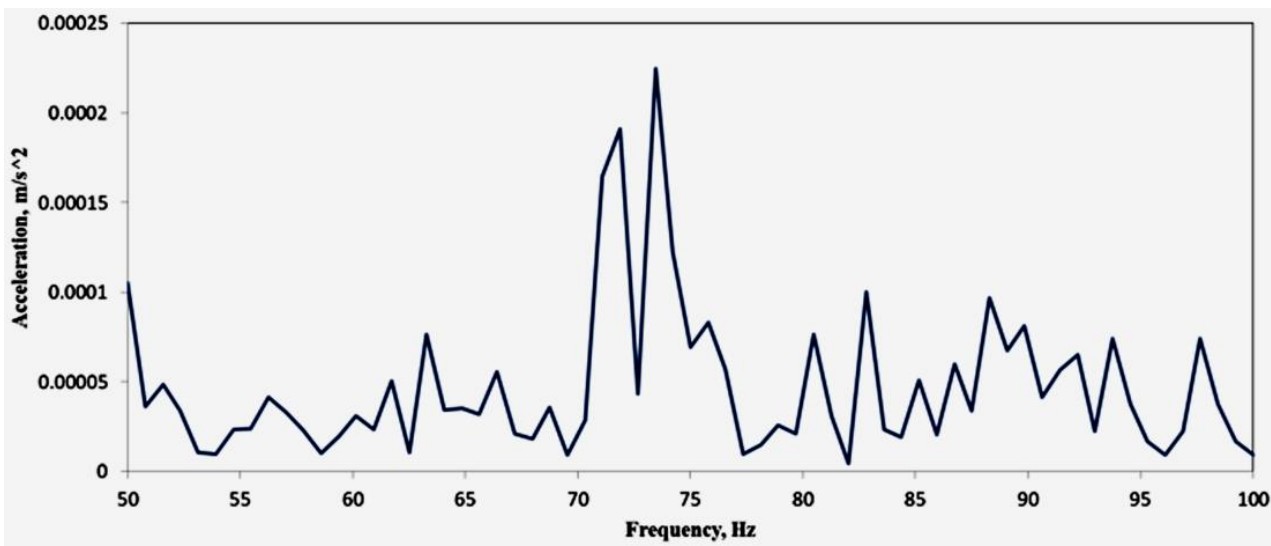

**Figure 10.** FRF of single crack blade.

For FRF spectrums, it was also observed that the amplitude or peak of the spectrum is different for different types of faults. The amplitude of vibration for the healthy blades was found to be 0.00018 m/s$^2$, as shown in Figure 9. For the faulty conditions such as single crack blade, two cracks in adjacent blades and mechanical looseness, the amplitude of vibrations were 0.00022 m/s$^2$, 0.00045 m/s$^2$ and 0.00047 m/s$^2$, as seen in Figures 10–12, respectively. Therefore, it can be concluded that the intensity of the crack increases, the stiffness of the blade reduces and the amplitude of vibrations increases.

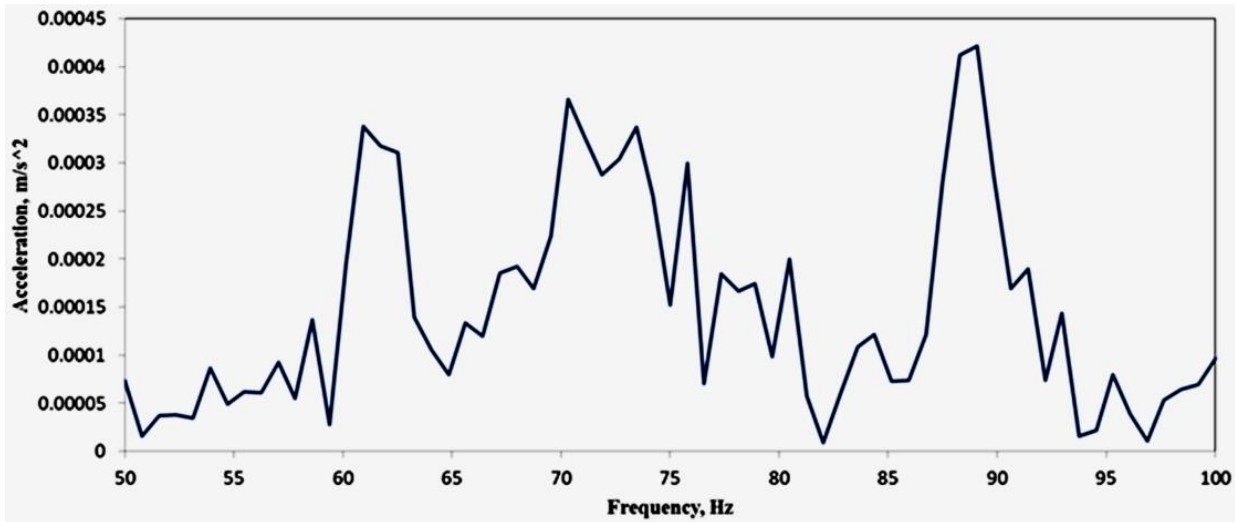

**Figure 11.** FRF of two cracks in adjacent blades.

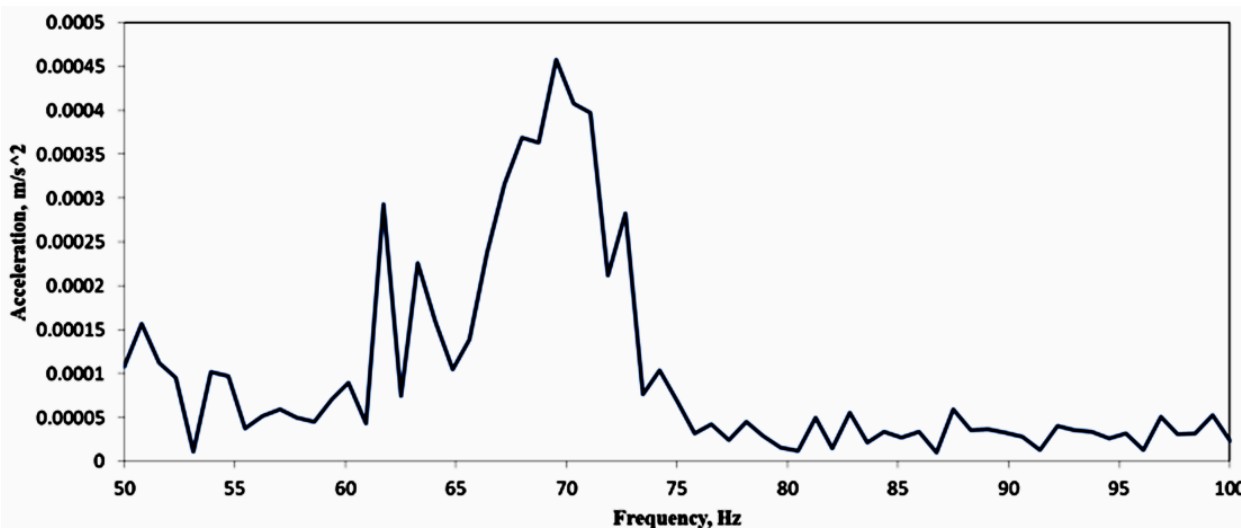

**Figure 12.** FRF of blade looseness.

It can be clearly seen from the results that every type of fault present in the rotating blades gives different FRF spectrums with multiple peaks within the resonance frequency range of the blade. These FRF spectrums are then used to develop a learning model using a support vector machine learning technique. This model proved to be the best in the classification of the faults compared with the other learning models. Hence, condition monitoring of the rotating blades is the most important activity that results in minimizing the costs incurred in maintenance and maximizing the safety of the equipment.

## 4. Conclusions

The most common failure cause of turbo machinery is blade faults. These faults may occur in many ways, but the most important ones are cracks and mechanical looseness of the blades. Therefore, it is necessary to detect these faults well in advance by providing a condition monitoring system in order to safeguard the machine. In this work, an IoT based fault diagnosis method was designed which included measuring the vibration data of the blades using a laser vibrometer sensor, a wireless sensor network system to transmit the data from the machine to the cloud platform and a support vector machine learning algorithm to predict the condition of the rotating blades. FRF spectrums were compared for

the healthy and faulty conditions of the rotating blades, and their features were extracted to help to classify the type of fault. The following are the key results obtained from the present work:

- The accuracy of the experimental setup was well obtained by verifying with simulation results;
- A remote fault detection method for rotating blades was developed by implementing the Internet of Things;
- A support vector machine algorithm was trained with vibration responses for healthy and faulty conditions of rotating blades with accuracy and precision;
- Fatigue cracks and mechanical looseness were detected using frequency response function spectrums.

This proposed work aims to provide an online condition monitoring method for Industry 4.0 applications.

**Author Contributions:** Conceptualization, R.P.B.K., M.K. and M.C.; methodology, R.P.B.K., M.K. and M.C.; software, R.P.B.K., M.K. and M.C.; validation, R.P.B.K., M.K. and M.C.; formal analysis, R.P.B.K., M.K. and M.C.; investigation, R.P.B.K., M.K. and M.C.; resources, R.P.B.K., M.K. and M.C.; data curation, R.P.B.K., M.K. and M.C.; writing—original draft preparation, R.P.B.K., M.K. and M.C.; writing—review and editing, M.C.; visualization, R.P.B.K., M.K. and M.C.; supervision, M.K.; project administration, R.P.B.K., M.K. and M.C. All authors have read and agreed to the published version of the manuscript.

**Funding:** This research received no external funding.

**Institutional Review Board Statement:** Not applicable.

**Informed Consent Statement:** Not applicable.

**Data Availability Statement:** Not applicable.

**Conflicts of Interest:** The authors declare no conflict of interest.

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
