# Peer review of "Real-Time Detection of Faults in Rotating Blades Using Frequency Response Function Analysis"

_2673-3161, doi:10.3390/applmech4010020_

Round 1

Reviewer 1 Report

The authors were tried to investigate the faults in rotating blades using real time detection system with frequency response function analysis. The authors were good tools to characterize the signal processing and defect detection.

I recommend the authors to strength few section by addressing the following minor comments.

1. Does the rotating blades has any limitation of temperature environments?

2. IS the defects/faults often occurs if the working environment is abnormal, evidence is needed.

3. Where is the application of blades and what is the purpose of using circular blades.

4. The function of sensor is to detect the rotations of the blades or performance of the blades?

5. What kind if faults it can be detected, is it possible to identify micro cracks?

6. The intensity of the cracks can be estimated or not using peak and valley of the curves.

7. Conclusion should be improved by providing the key results.

Author Response

## -- Reviewer-#1

S. No.

Comment

Answer

1

Does the rotating blades has any limitation of temperature environments?

In certain application of rotating blades like thermal power plants and aerospace jet engines, turbine blades are subjected to high temperatures along with centrifugal loads. But as these blades are coated with thermal resistant coatings can able to withstand thermal loads. Also, the effect of the elevated temperatures on the blades is appreciable only when the blade material has crystalline defects like dislocations. So, temperature environment is not considered in this work and is assumed that faults are developed purely due to mechanical loads.

2

Is the defects/faults often occurs if the working environment is abnormal, evidence is needed.

The abnormal condition of blade is found by using Campbell diagram from which the critical speed of the blades were identified comparing the excitation and natural frequencies of the rotating blades. Whenever the rotating blades operate at the resonance condition crack will initiate due to the higher stresses developed at the stress concentrated region of the blade. These stresses keep on increasing and the crack propagates till the point where the stress intensity factor of the crack exceeds the fracture toughness of the blade material before catastrophic failure occurs [17] That is why stress intensity factors are to be estimated for the propagating crack lengths based upon the linear elastic fracture mechanics For the above condition of the failure, It is seen in the literature that the final crack length is estimated to be 70% of the width of the blade which means that the blade does not fail at the crack length below final crack length. So, cracks should be detected before they reach final length.

17. Ravi Prakash Babu, K; Raghu Kumar, B; Devaraju, A; Murahari, K; Satyanarayana, K; Sai Kumar, G. Finite element modelling aspects in the fracture assessment of a low pressure steam turbine blade. International Journal of Interactive Design and Manufacturing 2022. Online. https://doi.org/10.1007/s12008-022-01045-2.

3

Where is the application of blades and what is the purpose of using circular blades.

These blades convert some form of energy into mechanical power. For example, wind turbine converts wind energy, when passed over the blades, into mechanical power by developing aerodynamic force. Likewise, steam turbine blades make the shaft to rotate when high pressure steam passes over them. Also, equipment like draught fans, blowers and compressors make use the kinetic energy of blades in order to increase the fluid outlet pressure.

4

The function of sensor is to detect the rotations of the blades or performance of the blades?

The sensor used in this work is laser vibrometer. Its function is to measure the vibration signals of the healthy and faulty conditions in the blades. These signals will help in understanding the frequency response function spectrum for each condition of blades so as detect faults in real time applications.

5

What kind if faults it can be detected, is it possible to identify micro cracks?

Vibrations are produced in the blades due to the faults namely crack, mechanical looseness, imbalance, misalignment etc. That means vibration data can be used to detect such faults. Generally, faults like imbalance and misalignment are monitored regularly during preventive maintenance works but the cracks and mechanical looseness of the blades becomes a challenging task to be found out. So, it becomes necessary to monitor the condition of the blades in order to detect cracks and mechanical looseness during the operation of the machine.

6

The intensity of the cracks can be estimated or not using peak and valley of the curves.

From FRF spectrums it is also observed that the amplitude or peak of the spectrum is different for different types of the faults. The amplitude of vibration for the healthy blades is found to be 0.00018 m/s2 as shown in Fig. 9. For the faulty conditions like single crack blade, two cracks in adjacent blades and mechanical looseness the amplitude of vibrations are 0.00022 m/s2, 0.00045 m/s2 and 0.00047 m/s2 as seen in Figs. 10, 11 and 12 respectively. So, it can be concluded that the intensity of the crack increases, the stiffness of the blade reduces and the amplitude of vibrations increases.

7

Conclusion should be improved by providing the key results.

The following are the key results obtained from the present work:

·         The accuracy of the experimental setup is well obtained by verifying with simulation results.

·         A remote fault detection method for rotating blades is developed by implementing Internet of Things.

·         Support vector machine algorithm is trained with vibration responses for healthy and faulty conditions of rotating blades with accuracy and precision.

·         Fatigue cracks and mechanical looseness are detected using frequency response function spectrums.

Reviewer 2 Report

Dear authors,

Your paper is interesting and the topic presented follows current trends. The article, however, has many faults. Furthermore, some editing is required.
The research and its design are not fully clear and some significant problems were detected. The methods used were described but not in detail.
Therefore, some additional corrections must be done for the article to be published as professional scientific work.

List below:
1.    The article is not always clearly written and easy to follow.

2.    The authors give relevant references which are linked to their study. However, the number of references and especially evaluation in the introduction is limited and not fully presenting the state of the art of the field.

3.     The abstract is well written introducing the basic overview of the paper. It is also written in a way that even a person not familiar with the topic can understand what the authors are proposing in their research. The reviewer would suggest not including any abbreviations in the abstract for clarity reasons(FRF and the same time no abbreviation for IoT).

4.    The introduction requires some improvements. The purpose of the introduction is to present the problem of the article and clearly present the overview of the state of the art in case of the topic and presented later and methods. Some problems noticed:
a.    First paragraph- there are some definite statements with no connection to the literature.
b.    second paragraph- there are some definite statements with no connection to the literature.
c.    Your topic is partially connected to quality control but no information about this aspect
d.    No evaluation of equipement and mentods used for FRF aquirment. Especially since the authors use a non-contact way (vibrometer) it is suggested to include in the introduction examples of FRF obtained using contact and none contact methods. It is suggested to include acoustic-based inspections and optical ones like fast cameras and motion amplification (eg https://www.researchgate.net/publication/326881537_Motion_Amplification_Technology_as_a_Tool_to_Support_Maintenance_Decisions) and state of art vibrometers, especially 3D LDV  that can be connected straight with cloud solutions and act as IoT sensor or connected to FEM software for validation purposes- reference  DOI: 10.3390/s23031263. This will allow the authors to explain why they have chosen a vibrometer as a testing tool for the rotating object.
e.    In the last paragraph were aim and scope are presented, however, the novelty is not presented strongly enough (in the scope of state-of-art evaluation).

5.    Chapter 2 some comments:
a.    No data on the vibrometer parameters as well on the signal acquisition settings.  This is problematic in case anyone wants to replicate the results.
b.    This is a 2D- single-point LDV. Please include this information as it is a way to obtain a single FRF for the measurement point. Include limitations of such single pint measurement – no full modal analysis of the structure (in comparison with 3D LDV)
c.    Fig 3 please make it bigger so everything is visible
d.    Values and units – usually there should be a space between the value and the unit (except degC and %). Please check the whole article. Problem noticed eg. Line 139, 144
e.    Line 153 “… located in the Dynamics lab.” Please use the professional name laboratory and include where it is located.
f.    Chapter 2.2 has many statements with no references to literature. Similar to chapter 2.3.
g.    Fig. 4 please make it bigger and indicate the crack on the photo

6.    Results comments:
a.    Table 1 – are those results for the first mode? No information was given. Please extend the table and show at least 3 modes. Similar as it is in the figure below.
b.    Fig. 8 must be enlarged. No frequencies are given for any of the modes or their type etc.
c.     As the information about equipment and acquisition parameters was not given it is hardly possible to evaluate the results. From Fig.9-12 it looks like the resolutions were not very high.

7.    The authors have introduced abbreviations at the beginning of the article but do not use them later. The most visible example is using all the time the full name “frequency response function”

8.    Conclusions are acceptable. However, an emphasis on novelty would be profitable.  Especially, since many commercial solutions already allow for implementing vibrometers as IoT tools for condition monitoring using manufacturers' software and cloud solutions (E.g Polytec solutions).  How author's solution is different/better?

Conclusions
The article has the potential to be interesting but after corrections to some significant problems found in the research and with some questions that need to be answered. Some additional state-of-the-art analysis of modern modal analysis testing techniques has to be incorporated in the introduction. At the current stage, this is a borderline paper and the reviewer asks for major changes in those areas and will be happy to check the paper after those corrections.

Author Response

## -- Reviewer-#2

S. No.

Comment

Answer

1

The article is not always clearly written and easy to follow

Article is written to the best of the knowledge

2

The authors give relevant references which are linked to their study. However, the number of references and especially evaluation in the introduction is limited and not fully presenting the state of the art of the field.

A reference was added based on the recommendation given by the reviewer and introduction is also properly explained.

3

The abstract is well written introducing the basic overview of the paper. It is also written in a way that even a person not familiar with the topic can understand what the authors are proposing in their research. The reviewer would suggest not including any abbreviations in the abstract for clarity reasons (FRF and the same time no abbreviation for IoT).

Corrections were made as per the reviewer’s suggestion

4

The introduction requires some improvements. The purpose of the introduction is to present the problem of the article and clearly present the overview of the state of the art in case of the topic and presented later and methods. Some problems noticed:

a. First paragraph- there are some definite statements with no connection to the literature.

b. second paragraph- there are some definite statements with no connection to the literature.

c. Your topic is partially connected to quality control but no information about this aspect

d. No evaluation of equipment and methods used for FRF acquirement. Especially since the authors use a non-contact way (vibrometer) it is suggested to include in the introduction examples of FRF obtained using contact and none contact methods. It is suggested to include acoustic-based inspections and optical ones like fast cameras and motion amplification (eg https://www.researchgate.net/publication/326881537_Motion_Amplification_Technology_as_a_Tool_to_Support_Maintenance_Decisions) and state of art vibrometers, especially 3D LDV that can be connected straight with cloud solutions and act as IoT sensor or connected to FEM software for validation purposes- reference  DOI: 10.3390/s23031263. This will allow the authors to explain why they have chosen a vibrometer as a testing tool for the rotating object.

e. In the last paragraph were aim and scope are presented, however, the novelty is not presented strongly enough (in the scope of state-of-art evaluation).

a, b. First two paragraphs of the introduction enlightens the general information regarding the performance of rotating blades in present day industrial scenario.

c. Sometimes, rotating blades are to be monitored for defects developed during the manufacturing process or made by less qualified personnel. So, quality management is essential during product development along with condition monitoring during the performance stage for the blades. Any methodology developed for the condition monitoring can be made used for the quality assurance along with the quality control of the blades. The emerging technologies like internet of things and machine learning techniques along with cloud computing can be implemented in developing real time fault detection method that can achieve efficient quality management of the blades. In doing so, sensors play a vital role to measure and acquire the data from the rotating blades.

d. In recent years, condition monitoring of industrial equipment is carried out by making use of the concepts available in the emerging technologies like Industrial Internet of Things (IIoT) and the 4th Industrial Revolution. The sensors capable of acquiring the data in the form of images, graphics and digitalization are processed to identify the condition of the equipment [17].

17. Roque; A; Sobral, J..Motion Amplification Technology as a Tool to Support Maintenance Decisions. Coimbra 2018. 58-64.

e. The work presented in this paper is believed to be useful in finding the blade faults of the machinery operating at the remote places with minimum human intervention.

5

Chapter 2 some comments:

a.    No data on the vibrometer parameters as well on the signal acquisition settings.  This is problematic in case anyone wants to replicate the results.

b.    This is a 2D- single-point LDV. Please include this information as it is a way to obtain a single FRF for the measurement point. Include limitations of such single point measurement – no full modal analysis of the structure (in comparison with 3D LDV)

c.    Fig 3 please make it bigger so everything is visible

d.    Values and units – usually there should be a space between the value and the unit (except degC and %). Please check the whole article. Problem noticed eg. Line 139, 144

e.    Line 153 “… located in the Dynamics lab.” Please use the professional name laboratory and include where it is located.

f.    Chapter 2.2 has many statements with no references to literature. Similar to chapter 2.3.

g.    Fig. 4 please make it bigger and indicate the crack on the photo

a. Laser vibrometer is of Polytec make consisting of OFV511 compact sensor head along with a laser unit as shown in Fig. 3(b). This laser unit delivers a beam of helium neon laser light of 316 nm through an optical fiber and is received by a high precision interferometer available in the head. The laser beam is split by a beam splitter into two beams namely a reference beam and a measurement beam. The measurement beam is then focused onto the rotating blades by making to pass through second beam splitter. The reflected measurement beam is then merges with the reference beam by the third beam splitter and focused on the photo detector. A Doppler frequency shift of the measurement beam is manifested with the change in the optical path length per unit of time. An acousto-optic modulator placed in the reference beam creates the frequency shift and the modulation frequency is directly proportional to the velocity of the rotating blades.

b. Non-contact type sensors can reduce the use of a greater number of contact sensors to measure the detailed vibration response of a rotating blade. The non-contact type sensors like laser vibrometers are available in 1D or 3D based upon the detailed vibration measurement. 3D laser vibrometers provide more detailed vibration information compared to other vibrometers. So, the selection of vibrometer depends upon the exploring the amount of vibration details. For the fault detection of rotating blades 1D vibrometer information is sufficient enough to perform frequency response function analysis.

Corrections were made as per the reviewer’s suggestion

Corrections were made as per the reviewer’s suggestion

Corrections were made as per the reviewer’s suggestion

f. Modal testing and IoT are very known these days and are studied during academics itself. So, references are not given.

Corrections were made as per the reviewer’s suggestion

6

Results comments:

a.    Table 1 – are those results for the first mode? No information was given. Please extend the table and show at least 3 modes. Similar as it is in the figure below.

b.    Fig. 8 must be enlarged. No frequencies are given for any of the modes or their type etc.

c.     As the information about equipment and acquisition parameters was not given it is hardly possible to evaluate the results. From Fig.9-12 it looks like the resolutions were not very high.

Condition

1st mode Frequency (Hz)

2nd mode Frequency (Hz)

3rd mode Frequency (Hz)

Healthy

77.01

599.04

893.71

Single crack

74.78

596.8

886.2

Two cracks

74.92

594.6

878.1

Looseness

73.24

593.2

873.5

Corrections were made as per the reviewer’s suggestion

As per the GUI settings, the resolution is limited.

7

The authors have introduced abbreviations at the beginning of the article but do not use them later. The most visible example is using all the time the full name “frequency response function”

Corrections were made as per the reviewer’s suggestion

8

Conclusions are acceptable. However, an emphasis on novelty would be profitable.  Especially, since many commercial solutions already allow for implementing vibrometers as IoT tools for condition monitoring using manufacturers' software and cloud solutions (E.g Polytec solutions).  How author's solution is different/better?

The following are the key results obtained from the present work:

·         The accuracy of the experimental setup is well obtained by verifying with simulation results.

·         A remote fault detection method for rotating blades is developed by implementing Internet of Things.

·         Support vector machine algorithm is trained with vibration responses for healthy and faulty conditions of rotating blades with accuracy and precision.

·         Fatigue cracks and mechanical looseness are detected using frequency response function spectrums.

Round 2

Reviewer 2 Report

Dear Authors,

Although, the paper has still some elements to be improved like a limited literature review on the topic, at this time I do not have any further suggestions about the methodology and results presentation. The paper can be considered for publication.